# Reciprocal Symmetry via Inverse Series Pairs

Wenchang Chu 

School of Mathematics and Statistics, Zhoukou Normal University, Zhoukou 466001, China;
hypergeometricx@outlook.com or wenchang.chu@unisalento.it

**Abstract:** Reciprocal series are employed to systematically review convolution sums, orthogonality relations, recurrence relations and reciprocal formulae for several classical number sequences, such as binomial coefficients, Stirling numbers, Bernoulli numbers, and Euler numbers.

**Keywords:** formal power series; inverse series pair; reciprocal relations

## 1. Introduction and Outline

There are numerous important number/polynomial sequence pairs in the mathematical literature that are tied by symmetric reciprocities. By employing reciprocal function pairs and formal power series expansions, an overview about explicit expressions, recurrence relations, orthogonality relations, convolution sums, and reciprocal formulae will systematically be presented for several typical classical sequences, such as binomial coefficients, Stirling numbers, Bernoulli/Euler numbers, and polynomials.

The general framework for symmetric reciprocities about abstract sequences will be set in the next section. Then, applications will be given in the remaining sections. The contents are organized as follows:

- Background about formal power series;
- Binomial-related coefficients;
- The Stirling numbers;
- arcsin and multifold Euler sums;
- arctan and multiple zeta values;
- Composite series pairs;
- Further exploration.

## 2. Framework about Formal Power Series

In this section, we shall review some general results about the formal power series, their coefficients, and their associated linear relations, which constitute the basis for manipulating specific functions and sequences in the subsequent sections.

Denote by $[x^k]\phi(x)$ the coefficient of $x^k$ in the formal power series $\phi(x)$. We shall frequently utilize the Lagrange inversion theorem, which is fundamental in classical analysis and combinatorics computation.

**The Lagrange Inversion Theorem** [1] (see Comtet [2] (§3.8) and [3,4]). For a formal power series $\varphi(y)$ subject to the condition $\varphi(0) \neq 0$, the functional equation $x = y/\varphi(y)$ determines $y$ as an implicit function of $x$. Then, for another formal power series $F(y)$ in the variable $y$, the following expansions (called Lagrange expansion formulae) hold for both composite series:

$$F(y(x)) = F(0) + \sum_{n=1}^{\infty} \frac{x^n}{n} [y^{n-1}] \{ F'(y) \varphi^n(y) \},\tag{1}$$

$$\frac{F(y(x))}{1 - (y\varphi'(y)/\varphi(y))} = \sum_{n=0}^{\infty} x^n [y^n] \{ F(y) \varphi^n(y) \}.\tag{2}$$

### 2.1. Composite Series

Denote by $\langle \phi; k \rangle$ and $\lfloor \phi; k \rceil$ the respective coefficients of $x^k$ and $\frac{x^k}{k!}$ in the formal power series $\phi(x)$:

$$\phi(x) = \sum_{k=0}^{\infty} x^k \langle \phi; k \rangle = \sum_{k=0}^{\infty} \frac{x^k}{k!} \lfloor \phi; k \rceil \quad \text{with} \quad \lfloor \phi; k \rceil = k! \langle \phi; k \rangle.$$

Suppose that $\psi(x)$ is another series with $\psi(0) = 0$. Then, the composite series $\phi \circ \psi$ is also a formal power series of $x$. Denote by $\sigma_n(m)$ the set of partitions of $m$ into $n$ parts, represented by $[1^{k_1} 2^{k_2} \cdots m^{k_m}]$ subject to the conditions $\sum_{i=1}^{m} k_i = n$ and $\sum_{i=1}^{m} i k_i = m$. For the coefficients of $\psi(x)$, we define the ordinary and exponential partial Bell polynomials (see Comtet [2] (§3.3) and [5–8]), respectively, by

$$\mathbf{B}_{m,n}(\psi) = n! \sum_{\sigma_n(m)} \prod_{i=1}^{m} \frac{\langle \psi; i \rangle^{k_i}}{k_i!} \quad \text{and} \quad \mathbf{B}_{m,n}^{\star}(\psi) = m! \sum_{\sigma_n(m)} \prod_{i=1}^{m} \frac{\lfloor \psi; i \rceil^{k_i}}{k_i! i!^{k_i}}.$$

Then, the coefficients of the composite series $\phi \circ \psi(x)$

$$\phi \circ \psi(x) = \sum_{n=0}^{\infty} \psi^n(x) \langle \phi; n \rangle = \sum_{m=0}^{\infty} x^m \langle \phi \circ \psi; m \rangle,$$

$$\phi \circ \psi(x) = \sum_{n=0}^{\infty} \frac{\psi^n(x)}{n!} \lfloor \phi; n \rceil = \sum_{m=0}^{\infty} \frac{x^m}{m!} \lfloor \phi \circ \psi; m \rceil;$$

are expressed in terms of Bell polynomials

$$\langle \phi \circ \psi; m \rangle = \sum_{n=0}^{m} \langle \phi; n \rangle \langle \psi^n; m \rangle = \sum_{n=0}^{m} \langle \phi; n \rangle \mathbf{B}_{m,n}(\psi),\tag{3}$$

$$\lfloor \phi \circ \psi; m \rceil = \sum_{n=0}^{m} \frac{\lfloor \phi; n \rceil}{n!} \lfloor \psi^n; m \rceil = \sum_{n=0}^{m} \lfloor \phi; n \rceil \mathbf{B}_{m,n}^{\star}(\psi).\tag{4}$$

### 2.2. Univalent Series

A formal power series $f(x)$ is univalent if it satisfies the conditions $f(0) = 0$ and $f'(0) = 1$. Define the ordinary and exponential connection coefficients by

$$f^n(x) = \sum_{m=n}^{\infty} x^m \Phi(m, n) \quad \text{and} \quad \frac{f^n(x)}{n!} = \sum_{m=n}^{\infty} \frac{x^m}{m!} \Phi^{\star}(m, n).\tag{5}$$

They can obviously be expressed via each other

$$\Phi(m, n) = \langle f^n(x); m \rangle \text{ and } \Phi^{\star}(m, n) = \left\lfloor \frac{f^n(x)}{n!}; m \right\rceil = \frac{m!}{n!} \Phi(m, n).\tag{6}$$

Both $\Phi(m, n)$ and $\Phi^{\star}(m, n)$ admit the following useful properties.

- Expressions in Bell polynomials:

$$\Phi(m, n) = \mathbf{B}_{m,n}(f),\tag{7}$$

$$\Phi^{\star}(m, n) = \mathbf{B}_{m,n}^{\star}(f).\tag{8}$$

- Symmetric convolution:

$$\Phi(m, p+q) = \sum_{k=p}^{m-q} \Phi(k, p)\Phi(m-k, q), \tag{9}$$

$$\binom{p+q}{p}\Phi^\star(m, p+q) = \sum_{k=p}^{m-q} \binom{m}{k}\Phi^\star(k, p)\Phi^\star(m-k, q). \tag{10}$$

- Recurrence relations:

$$\Phi(m, n) = \sum_{k=0}^{m-n} \binom{n}{k}\binom{m-2n}{m-n-k}\Phi(m-n+k, k), \tag{11}$$

$$\Phi^\star(m, n) = \sum_{k=0}^{m-n} \binom{m}{n-k}\binom{m-2n}{m-n-k}\Phi^\star(m-n+k, k). \tag{12}$$

The first relation (11) can be shown as follows. Specifying $F(y) = y^n$ and $y = f(x)$ in (1) implies that the inverse function $x = g(y)$. Writing $g(y) = y/\varphi(y)$ with $\varphi(y) = y/g(y)$, we can carry out the following operation:

$$
\begin{aligned}
\Phi(m, n) = [x^m]f^n(x) &= [x^{m-n}]\left\{1 - \left(1 - \frac{f(x)}{x}\right)\right\}^n \\
&= [x^{m-n}]\sum_{j=0}^{m-n}(-1)^j\binom{n}{j}\left(1 - \frac{f(x)}{x}\right)^j \\
&= [x^{m-n}]\sum_{j=0}^{m-n}(-1)^j\binom{n}{j}\sum_{k=0}^{j}(-1)^k\binom{j}{k}\left(\frac{f(x)}{x}\right)^k \\
&= \sum_{k=0}^{m-n}\binom{n}{k}[x^{m-n+k}]f^k(x)\sum_{j=k}^{m-n}(-1)^{j-k}\binom{n-k}{j-k}.
\end{aligned}
$$

Then, the curious equality (11) follows by evaluating the last binomial sum

$$\sum_{j=k}^{m-n}(-1)^{j-k}\binom{n-k}{j-k} = \binom{m-2n}{m-n-k}.$$

Taking into account the binomial product

$$\frac{m!}{n!}\binom{n}{k} = \frac{(m-n+k)!}{k!}\binom{m}{n-k},$$

we deduce the exponential form displayed in (12), to which there is an equivalent form due to Sun [9].

We remark that both Formulas (11) and (12) make sense only for $n \le m < 2n$, because otherwise, we would have two trivial equalities for $m < n$ and $m \ge 2n$. Under the replacement $m \to m + n$, two recurrence relations (11) and (12) can be restated equivalently as the following self-reciprocal relations:

$$\Phi(m+n, n) = \sum_{k=0}^{n}\binom{n}{k}\binom{m-n}{m-k}\Phi(m+k, k), \tag{13}$$

$$\Phi^\star(m+n, n) = \sum_{k=0}^{n}\binom{m+n}{m+k}\binom{m-n}{m-k}\Phi^\star(m+k, k). \tag{14}$$

*2.3. Reciprocal Series*

When $g(x)$ is another univalent series with $g(0) = 0$ and $g'(0) = 1$, its ordinary and exponential connection coefficients are given by

$$g^n(x) = \sum_{m=n}^{\infty} x^m \Psi(m,n) \quad \text{and} \quad \frac{g^n(x)}{n!} = \sum_{m=n}^{\infty} \frac{x^m}{m!} \Psi^\star(m,n) \tag{15}$$

which can be expressed via each other

$$\Psi(m,n) = [x^m]g^n(x) \quad \text{and} \quad \Psi^\star(m,n) = \frac{m!}{n!}\Psi(m,n). \tag{16}$$

The two series $f(x)$ and $g(x)$ are called "compositional reciprocal" if and only if $f(g(x)) = g(f(x)) = x$, i.e., one is the compositional inverse of the other. If $\{f,g\}$ and $\{F,G\}$ are two reciprocal pairs, then both $\{-f(-x), -g(-x)\}$ and $\{F(f), g(G)\}$ are again reciprocal pairs. Supposing that $f(x)$ and $g(x)$ are reciprocal univalent series with their coefficients being given by (5) and (15), we have the following orthogonality relations

$$\begin{aligned} \sum_{k=n}^{m} \Phi(m,k)\Psi(k,n) &= \chi(m=n), \\ \sum_{k=n}^{m} \Psi(m,k)\Phi(k,n) &= \chi(m=n); \end{aligned} \tag{17}$$

where $\chi$ stands for the logical function with $\chi(\text{true}) = 1$ and $\chi(\text{false}) = 0$. They can be verified by comparing the coefficients of $y^m$ in the following equations:

$$\begin{aligned} y^n = g^n(f(y)) &= \sum_{k=n}^{\infty} f^k(y)\Psi(k,n) \\ &= \sum_{k=n}^{\infty} \Psi(k,n) \sum_{m=k}^{\infty} \Phi(m,k)y^m \\ &= \sum_{m=n}^{\infty} y^m \sum_{k=n}^{m} \Phi(m,k)\Psi(k,n). \end{aligned}$$

Expressing in terms of the exponential coefficients, the orthogonality relations have the same form:

$$\begin{aligned} \sum_{k=n}^{m} \Phi^\star(m,k)\Psi^\star(k,n) &= \chi(m=n), \\ \sum_{k=n}^{m} \Psi^\star(m,k)\Phi^\star(k,n) &= \chi(m=n). \end{aligned} \tag{18}$$

Let $F(y) = y^n$ and $y = f(x)$ as before. This implies again that $x = g(y) = y/\varphi(y)$ with $\varphi(y) = y/g(y)$. According to (1), we have

$$\Phi(m,n) = [x^m]f^n(x) = [x^m]y^n = \frac{n}{m}[y^{m-1}]\frac{y^{m+n-1}}{g^m(y)}.$$

This leads us to the following alternative expressions:

$$\Phi(m,n) = \frac{n}{m}[y^{m-n}]\frac{y^m}{g^m(y)} \quad \text{and} \quad \Phi^\star(m,n) = \frac{(m-1)!}{(n-1)!}[y^{m-n}]\frac{y^m}{g^m(y)}, \tag{19}$$

$$\Psi(m,n) = \frac{n}{m}[y^{m-n}]\frac{y^m}{f^m(y)} \quad \text{and} \quad \Psi^\star(m,n) = \frac{(m-1)!}{(n-1)!}[y^{m-n}]\frac{y^m}{f^m(y)}. \tag{20}$$

The first expression (19) can be manipulated further as follows:

$$
\begin{aligned}
\Phi(m,n) &= \frac{n}{m}[y^{m-n}]\frac{y^m}{g^m(y)} = \frac{n}{m}[y^{m-n}]\left\{1 - \left(1 - \frac{g(y)}{y}\right)\right\}^{-m} \\
&= \frac{n}{m}[y^{m-n}]\sum_{j=0}^{m-n}(-1)^j\binom{-m}{j}\left(1 - \frac{g(y)}{y}\right)^j \\
&= \frac{n}{m}[y^{m-n}]\sum_{j=0}^{m-n}(-1)^j\binom{-m}{j}\sum_{k=0}^{j}(-1)^k\binom{j}{k}\left(\frac{g(y)}{y}\right)^k \\
&= \frac{n}{m}\sum_{k=0}^{m-n}\binom{-m}{k}[y^{m-n+k}]g^k(y)\sum_{j=k}^{m-n}(-1)^{j-k}\binom{-m-k}{j-k}.
\end{aligned}
$$

Evaluating the last binomial sum

$$
\sum_{j=k}^{m-n}(-1)^{j-k}\binom{-m-k}{j-k} = \binom{2m-n}{m+k}
$$

we derive the following reciprocally symmetric relations:

$$
\begin{aligned}
\Phi(m,n) &= \frac{n}{m}\sum_{k=0}^{m-n}\binom{-m}{k}\binom{2m-n}{m+k}\Psi(m-n+k,k), \\
\Psi(m,n) &= \frac{n}{m}\sum_{k=0}^{m-n}\binom{-m}{k}\binom{2m-n}{m+k}\Phi(m-n+k,k).
\end{aligned}
\tag{21}
$$

Under the replacement $m \to m+n$, they can be restated as the reciprocities below

$$
\begin{aligned}
\Phi(m+n,n) &= \binom{2m+n}{m}\sum_{k=0}^{m}\frac{n(-1)^k}{m+n+k}\binom{m}{k}\Psi(m+k,k), \\
\Psi(m+n,n) &= \binom{2m+n}{m}\sum_{k=0}^{m}\frac{n(-1)^k}{m+n+k}\binom{m}{k}\Phi(m+k,k).
\end{aligned}
\tag{22}
$$

Keeping in mind the binomial product

$$
\frac{m!n}{n!m}\binom{-m}{k} = (-1)^k\frac{(m-n+k)!}{k!}\binom{m+k-1}{n-1},
$$

we recover the symmetric reciprocities due to Hsu [10]:

$$
\begin{aligned}
\Phi^\star(m,n) &= \sum_{k=0}^{m-n}(-1)^k\binom{2m-n}{m+k}\binom{m+k-1}{n-1}\Psi^\star(m-n+k,k), \\
\Psi^\star(m,n) &= \sum_{k=0}^{m-n}(-1)^k\binom{2m-n}{m+k}\binom{m+k-1}{n-1}\Phi^\star(m-n+k,k).
\end{aligned}
\tag{23}
$$

By replacing $m$ by $m+n$, we can further restate the above relations as

$$
\begin{aligned}
\Phi^\star(m+n,n) &= \sum_{k=0}^{m}(-1)^m\binom{-n}{m+k}\binom{2m+n}{m-k}\Psi^\star(m+k,k), \\
\Psi^\star(m+n,n) &= \sum_{k=0}^{m}(-1)^m\binom{-n}{m+k}\binom{2m+n}{m-k}\Phi^\star(m+k,k).
\end{aligned}
\tag{24}
$$

Alternatively, the above relations can be expressed as

$$\Phi^{\star}(m+n,n) = \binom{2m+n}{n} \sum_{k=0}^{m} \frac{n(-1)^k}{m+n+k} \binom{2m}{m+k} \Psi^{\star}(m+k,k),$$

$$\Psi^{\star}(m+n,n) = \binom{2m+n}{n} \sum_{k=0}^{m} \frac{n(-1)^k}{m+n+k} \binom{2m}{m+k} \Phi^{\star}(m+k,k).$$

(25)

They are equivalent to the orthogonality relation:

$$\sum_{k=0}^{m} \binom{-n}{m+k} \binom{2m+n}{m-k} \binom{-k}{m+\ell} \binom{2m+k}{m-\ell} = \binom{m+n}{m+\ell} \binom{m-n}{m-\ell} = \delta_{n,\ell}.$$

For $n \in \mathbb{N}_0$ and an indeterminate $x$, define the rising and falling factorials by $(x)_0 = \langle x \rangle_0 = 1$ and

$$\left. \begin{array}{l} (x)_n = x(x+1)\cdots(x+n-1) \\ \langle x \rangle_n = x(x-1)\cdots(x-n+1) \end{array} \right\} \quad \text{for} \quad x \in \mathbb{N}.$$

In terms of classical hypergeometric series (see Bailey [11] §2.5)

$$_pF_q \left[ \begin{array}{cccc} a_1, & a_2, & \cdots, & a_p \\ b_1, & b_2, & \cdots, & b_q \end{array} \middle| z \right] = \sum_{k=0}^{\infty} \frac{(a_1)_k (a_2)_k \cdots (a_p)_k}{(b_1)_k (b_2)_k \cdots (b_q)_k} \frac{z^k}{k!},$$

the last sum can be expressed as a terminating balanced $_4F_3$-series

$$_4F_3 \left[ \begin{array}{c} 1-m, m+n+1, m+\ell+1, 2m+2 \\ m+n+2, m+\ell+2, \ m+2 \end{array} \middle| 1 \right]$$

$$= (-1)^m \frac{(m+n+1)(m+\ell+1)(n-m)_m}{m(\ell-n)(m+n+1)_m \binom{2m+1}{m}}.$$

The above identity admits the following extension for the balanced $_4F_3$-series:

$$_4F_3 \left[ \begin{array}{cccc} a, & b, & d, & -n \\ 1+b, 1+d, a-n \end{array} \middle| 1 \right] = \frac{n!}{(1-a)_n} \left\{ \frac{b(1+d-a)_n}{(b-d)(1+d)_n} - \frac{d(1+b-a)_n}{(b-d)(1+b)_n} \right\}.$$

This can be verified by combining the partial fractions

$$\frac{bd}{(b+k)(d+k)} = \frac{bd}{(b-d)(d+k)} + \frac{bd}{(d-b)(b+k)}$$

with the Pfaff–Saalschütz summation theorem (see Bailey [11] §2.2)

$$_3F_2 \left[ \begin{array}{ccc} a, & b, & -n \\ c, 1+a+b-c-n \end{array} \middle| 1 \right] = \frac{(c-a)_n (c-b)_n}{(c)_n (c-a-b)_n}.$$

## 3. Binomial-Related Coefficients

For a formal power series $\varphi(x)$ with $\varphi(0) = 1$, we can define a univalent function by $f(x) = x/\varphi(x)$, which determines the implicit inverse function $g(x) = f^{-1}(x)$. For this reciprocal pair $\{f, g\}$, recall the connection coefficients (19) and (20):

$$\Phi(m,n) = [x^m] f^n(x) = [x^{m-n}] \varphi^{-n}(x),$$

$$\Psi(m,n) = [x^m] g^n(x) = \frac{n}{m} [x^{m-n}] \varphi^m(x).$$

By choosing properly $\varphi(x)$, we are going to examine five classes of reciprocal relations related to binomial coefficients. We shall confine ourselves to highlighting the formu-

lae concerning ordinary coefficients $\{\Phi(m,n), \Psi(m,n)\}$ without producing those about exponential ones $\{\Phi^\star(m,n), \Psi^\star(m,n)\}$.

*3.1. The Abel Coefficients*

By letting $\varphi(x) = e^{\lambda x}$, we obtain the Abel coefficients:

$$\Phi(m,n) = \frac{(-n\lambda)^{m-n}}{(m-n)!} \quad \text{and} \quad \Psi(m,n) = \frac{n}{m}\frac{(m\lambda)^{m-n}}{(m-n)!}.$$

Then, we deduce the following known results (cf. [3,12–14] §1.5).

- Orthogonality relations corresponding to (17):

$$\chi(m=n) = \sum_{k=n}^{m} \frac{k}{m}\frac{(m\lambda)^{m-k}}{(m-k)!}\frac{(-n\lambda)^{k-n}}{(k-n)!},$$

$$\chi(m=n) = \sum_{k=n}^{m} \frac{n}{k}\frac{(-k\lambda)^{m-k}}{(m-k)!}\frac{(k\lambda)^{k-n}}{(k-n)!},$$

where the first one is easy to check directly, while the second is not.

- Convolution formulae corresponding to (9):

$$\begin{aligned}\sum_{k=p}^{m-q} \frac{(-p)^{k-p}}{(k-p)!}\frac{(-q)^{m-k-q}}{(m-k-q)!} &= \frac{(-p-q)^{m-p-q}}{(m-p-q)!}, \\ \sum_{k=p}^{m-q} \frac{m}{k(m-k)}\frac{k^{k-p}}{(k-p)!}\frac{(m-k)^{m-k-q}}{(m-k-q)!} &= \frac{p+q}{pq}\frac{m^{m-p-q}}{(m-p-q)!}.\end{aligned} \tag{26}$$

- Recurrence relations corresponding to (11):

$$n^{m-n} = \sum_{k=0}^{m-n} \binom{n}{k}\binom{m-2n}{m-n-k}k^{m-n}, \tag{27}$$

$$\frac{n}{m}n^{m-n} = \sum_{k=0}^{m-n} \binom{n}{k}\binom{m-2n}{m-n-k}\frac{k}{m-n+k}(m-n+k)^{m-n}. \tag{28}$$

- Reciprocal relations corresponding to (21):

$$\frac{m}{n}(-n)^{m-n} = \sum_{k=0}^{m-n} \binom{-m}{k}\binom{2m-n}{m+k}\frac{k}{m-n+k}(m-n+k)^{m-n}, \tag{29}$$

$$m^{m-n} = \sum_{k=0}^{m-n} \binom{-m}{k}\binom{2m-n}{m+k}(-k)^{m-n}. \tag{30}$$

*3.2. The Gould Coefficients*

The Gould coefficients are defined via $\varphi(x) = (1+\beta x)^\lambda$ as follows:

$$\Phi(m,n) = \binom{-n\lambda}{m-n}\beta^{m-n} \quad \text{and} \quad \Psi(m,n) = \frac{n}{m}\binom{m\lambda}{m-n}\beta^{m-n}.$$

The following known results can be deduced (cf. [4,15–17]).

- Orthogonality relations corresponding to (17):

$$\chi(m = n) = \sum_{k=n}^{m} \frac{k}{m} \binom{m\lambda}{m-k} \binom{-n\lambda}{k-n},$$

$$\chi(m = n) = \sum_{k=n}^{m} \frac{n}{k} \binom{-k\lambda}{m-k} \binom{k\lambda}{k-n}.$$

- Convolution formulae corresponding to (9):

$$\sum_{k=p}^{m-q} \binom{-p\lambda}{k-p} \binom{-q\lambda}{m-k-q} = \binom{-(p+q)\lambda}{m-p-q},$$

$$\sum_{k=p}^{m-q} \frac{m}{k(m-k)} \binom{k\lambda}{k-p} \binom{(m-k)\lambda}{m-k-q} = \frac{p+q}{pq} \binom{m\lambda}{m-p-q}. \tag{31}$$

- Recurrence relations corresponding to (11):

$$\binom{-n\lambda}{m-n} = \sum_{k=0}^{m-n} \binom{n}{k} \binom{m-2n}{m-n-k} \binom{-k\lambda}{m-n}, \tag{32}$$

$$\frac{n}{m} \binom{m\lambda}{m-n} = \sum_{k=0}^{m-n} \binom{n}{k} \binom{m-2n}{m-n-k} \frac{k}{m-n+k} \binom{(m-n+k)\lambda}{m-n}. \tag{33}$$

- Reciprocal relations corresponding to (21):

$$\frac{m}{n} \binom{-n\lambda}{m-n} = \sum_{k=0}^{m-n} \binom{-m}{k} \binom{2m-n}{m+k} \frac{k}{m-n+k} \binom{(m-n+k)\lambda}{m-n}, \tag{34}$$

$$\binom{m\lambda}{m-n} = \sum_{k=0}^{m-n} \binom{-m}{k} \binom{2m-n}{m+k} \binom{-k\lambda}{m-n}. \tag{35}$$

In what follows, we briefly review Catalan numbers, Lah numbers, and binomial coefficients. For these connection coefficients, we shall not produce the formulae for the corresponding convolution (9), recurrence relations (11), orthogonality relations (17), and reciprocal relations (21), as the reader can write them down without difficulty.

### 3.3. Catalan Numbers

For $\varphi(x) = (1-x)^{-1}$, we have the inverse pair

$$f(x) = x(1-x) \quad \text{and} \quad g(x) = \frac{1 - \sqrt{1-4x}}{2}$$

where $\frac{g(x)}{x}$ is the generating function of the Catalan numbers (cf. [5,18–20]), and

$$\frac{1 - \sqrt{1-4x}}{2x} = \sum_{n=0}^{\infty} \frac{1}{n+1} \binom{2n}{n} x^n.$$

The connection coefficients (see Wilf [21] §2.5) are given explicitly by

$$\Phi(m+n, n) = (-1)^m \binom{n}{m} \quad \text{and} \quad \Psi(m+n, n) = \frac{n}{2m+n} \binom{2m+n}{m}.$$

### 3.4. Lah Numbers

Lah numbers are self-reciprocal, defined by the connection coefficients expressing rising factorials in terms of falling factorials (cf. [22], p. 44)

$$(x)_n = \sum_{k=1}^{n} (-1)^k L(n,k)(-x)_k \quad \text{and} \quad (-x)_n = \sum_{k=1}^{n} (-1)^k L(n,k)(x)_k.$$

They have the exponential generating function (equivalent to $\varphi(x) = 1 - x$)

$$\frac{1}{k!}\left(\frac{x}{1-x}\right)^k = \sum_{n=k}^{\infty} \frac{x^n}{n!} L(n,k)$$

and the explicit formula

$$L(n,k) = \frac{n!}{k!}\binom{n-1}{k-1}.$$

Denote by $S_1(m,k)$ and $S_2(m,k)$ (see Section 4) the Stirling numbers of the first and the second kinds, respectively. By manipulating the expressions

$$(-x)_m = (-1)^m \langle x \rangle_m = \sum_{k=0}^{m} (-1)^{m+k} S_1(m,k)(-x)^k$$

$$= \sum_{k=0}^{m} (-1)^{m+k} S_1(m,k) \sum_{n=0}^{k} \langle -x \rangle_n S_2(k,n)$$

$$= \sum_{n=0}^{m} (x)_n (-1)^{m+n} \sum_{k=n}^{m} (-1)^k S_1(m,k) S_2(k,n),$$

we recover the following known formula:

$$L(m,n) = \sum_{k=n}^{m} (-1)^{m+k} S_1(m,k) S_2(k,n).$$

More convolutions of a similar type have been evaluated in [23] (§6.1) and [24,25].

### 3.5. Binomial Coefficients

For $\varphi(x) = \sqrt{1+2x}$, we have the inverse pair:

$$f(x) = \frac{x}{\sqrt{1+2x}} \quad \text{and} \quad g(x) = f^{-1}(x) = x\left(x + \sqrt{1+x^2}\right).$$

The connection coefficients are given explicitly by

$$\Phi(m+n,n) = \binom{-\frac{n}{2}}{m} 2^m \quad \text{and} \quad \Psi(m+n,n) = \frac{n \cdot 2^m}{m+n}\binom{\frac{m+n}{2}}{m}. \tag{36}$$

Alternatively, for $\varphi(x) = \frac{2(\sqrt{1+x}-1)}{x} = \frac{2}{1+\sqrt{1+x}}$, we have another inverse pair, i.e.,

$$f(x) = \frac{x}{2}\left(1 + \sqrt{1+x}\right) \text{ and } g(x) = f^{-1}(x) = \frac{2(9x^2 + \sqrt{48x^3 + 81x^4})^{2/3} - 4 \cdot 6^{1/3}x}{6^{2/3}(9x^2 + \sqrt{48x^3 + 81x^4})^{1/3}}.$$

Even though the inverse function $g(x)$ looks ugly, the related connection coefficients are quite elegant, as shown below

$$\Phi(m+n,n) = \frac{n \cdot 4^{-m}}{n-m}\binom{n-m}{m} \text{ and } \Psi(m+n,n) = \frac{n(-1)^{m+n}}{m \cdot 4^m}\binom{-m}{2m+n}. \tag{37}$$

## 4. The Stirling Numbers

The Stirling numbers are defined by the connection coefficients (cf. [26–29])

$$\langle x \rangle_n = \sum_{k=0}^{n} x^k S_1(n,k) \quad \text{and} \quad x^n = \sum_{k=0}^{n} \langle x \rangle_k S_2(n,k)$$

with the following exponential generating functions:

$$\frac{\ln^n(1+x)}{n!} = \sum_{m=n}^{\infty} \frac{S_1(m,n)}{m!} x^m \quad \text{and} \quad \frac{(e^x - 1)^n}{n!} = \sum_{m=n}^{\infty} \frac{S_2(m,n)}{m!} x^m.$$

Euler's formula for the Stirling numbers of the second kind is as follows:

$$S_2(m,n) = \frac{1}{n!} \sum_{k=0}^{n} (-1)^k \binom{n}{k} (n-k)^m.$$

The symmetric formulae on reciprocities due to Schläfli–Schlömilch [30,31] (see also Carlitz [32] and Gould [33]) follow directly from (23)

$$S_1(m+n,n) = \sum_{k=0}^{m} (-1)^k \binom{2m+n}{m+n+k} \binom{m+n+k-1}{n-1} S_2(m+k,k), \tag{38}$$

$$S_2(m+n,n) = \sum_{k=0}^{m} (-1)^k \binom{2m+n}{m+n+k} \binom{m+n+k-1}{n-1} S_1(m+k,k). \tag{39}$$

The recurrence relations of Sun [9] follow from (14)

$$S_1(m+n,n) = \sum_{k=0}^{m} \binom{m+n}{n-k} \binom{m-n}{m-k} S_1(m+k,k), \tag{40}$$

$$S_2(m+n,n) = \sum_{k=0}^{m} \binom{m+n}{n-k} \binom{m-n}{m-k} S_2(m+k,k). \tag{41}$$

For an extra indeterminate $q$, define the $q$-binomial coefficients by

$$\begin{bmatrix} n \\ 0 \end{bmatrix} = 1 \quad \text{and} \quad \begin{bmatrix} n \\ k \end{bmatrix} = \prod_{j=1}^{k} \frac{1 - q^{n-j+1}}{1 - q^j} \quad \text{for} \quad k \in \mathbb{N}.$$

M. Josuat-Vergès [34] introduced the $q$-analogues of the Stirling numbers below

$$\mathcal{S}_1[m,n] = \sum_{j=0}^{m-n} \sum_{i=j}^{m-n} (-1)^i \frac{i+j+1}{m+j} \binom{m-i-2}{n-2} \binom{m+j}{n-1} \begin{bmatrix} i \\ j \end{bmatrix} \frac{q^{\binom{j+1}{2}}}{(1-q)^{m-n}}, \tag{42}$$

$$\mathcal{S}_2[m,n] = \sum_{j=0}^{n} \sum_{i=j}^{m-n} (-1)^i \frac{i+j+1}{m+1} \binom{m+1}{n+i+1} \binom{m+1}{n-j} \begin{bmatrix} i \\ j \end{bmatrix} \frac{q^{\binom{j+1}{2}}}{(1-q)^{m-n}}; \tag{43}$$

where the binomial coefficient $\binom{m}{n}$ is modified by two initial values $\binom{-1}{-1} = \binom{-2}{-2} = 1$.

They satisfy both the recurrence relation (12)

$$\mathcal{S}_1[m+n,n] = \sum_{k=0}^{m} \binom{m+n}{m+k} \binom{m-n}{m-k} \mathcal{S}_1[m+k,k], \tag{44}$$

$$\mathcal{S}_2[m+n,n] = \sum_{k=0}^{m} \binom{m+n}{m+k} \binom{m-n}{m-k} \mathcal{S}_2[m+k,k]; \tag{45}$$

and the reciprocal relation (23) with the exception of alternating signs

$$\mathcal{S}_1[m+n,n] = \sum_{k=0}^{m}(-1)^{m+k}\binom{m+n+k-1}{m+k}\binom{2m+n}{m-k}\mathcal{S}_2[m+k,k], \tag{46}$$

$$\mathcal{S}_2[m+n,n] = \sum_{k=0}^{m}(-1)^{m+k}\binom{m+n+k-1}{m+k}\binom{2m+n}{m-k}\mathcal{S}_1[m+k,k]. \tag{47}$$

However, their exponential generating functions are not reciprocal ones as (5) and (15). In fact, the author failed to determine them explicitly.

## 5. arcsin and Multifold Euler Sums

This section will be devoted to examining the reciprocal function pair sin and arcsin. Define the connection coefficients by exponential generating functions as follows

$$\begin{aligned}\frac{(\sin x)^n}{n!} &= \sum_{m=0}^{\infty}\frac{x^{2m+n}}{(2m+n)!}\mathrm{V}(2m+n,n),\\ \frac{(\arcsin x)^n}{n!} &= \sum_{m=0}^{\infty}\frac{x^{2m+n}}{(2m+n)!}\mathcal{V}(2m+n,n);\end{aligned} \tag{48}$$

which can be expressed as

$$\begin{aligned}\mathrm{V}(2m+n,n) &= \frac{(2m+n)!}{n!}[x^{2m+n}](\sin x)^n,\\ \mathcal{V}(2m+n,n) &= \frac{(2m+n)!}{n!}[x^{2m+n}](\arcsin x)^n.\end{aligned} \tag{49}$$

It is routine to check for all the $m, n \in \mathbb{N}$, the following values

$$\mathrm{V}(2m+n-1,n) = \mathcal{V}(2m+n-1,n) = 0.$$

and

$$(-1)^m\mathrm{V}(2m+n,n),\ \mathcal{V}(2m+n,n) \in \mathbb{N}.$$

*5.1. Explicit Formula of* **V**(*m*, *n*)

By means of the binomial theorem, we can manipulate the generating function

$$\begin{aligned}\mathrm{V}(2m+n,n) &= \frac{(2m+n)!}{n!}[x^{2m+n}](\sin x)^n\\ &= \frac{(2m+n)!}{n!}[x^{2m+n}]\left\{\frac{e^{x\sqrt{-1}}-e^{-x\sqrt{-1}}}{2\sqrt{-1}}\right\}^n\\ &= \frac{(2m+n)!}{n!}\sum_{k=0}^{n}\frac{(-1)^k}{(2\sqrt{-1})^n}\binom{n}{k}[x^{2m+n}]e^{(n-2k)x\sqrt{-1}},\end{aligned}$$

which results in the explicit formula

$$\mathrm{V}(2m+n,n) = \sum_{k=0}^{n}(-1)^{m+k}\binom{n}{k}\frac{(n-2k)^{2m+n}}{2^n\cdot n!}. \tag{50}$$

*5.2. Explicit Formula of $\mathcal{V}(m,n)$*

According to (24), we have the reciprocal formulae:

$$\mathcal{V}(2m+n,n) = \sum_{k=0}^{2m} \binom{4m+n}{2m-k}\binom{-n}{2m+k} V(2m+k,k),$$

$$V(2m+n,n) = \sum_{k=0}^{2m} \binom{4m+n}{2m-k}\binom{-n}{2m+k} \mathcal{V}(2m+k,k).$$

(51)

Instead, there exist the following shorter relations:

$$\mathcal{V}(2m+n,n) = \sum_{k=0}^{m} \binom{3m+n}{m-k}\binom{-n}{2m+k} V(2m+k,k),$$

$$V(2m+n,n) = \sum_{k=0}^{m} \binom{3m+n}{m-k}\binom{-n}{2m+k} \mathcal{V}(2m+k,k).$$

(52)

The first relation is shown below and the second one can be done similarly. By specifying $f(y) = \sin y$ in (20), we obtain

$$\mathcal{V}(2m+n,n) = \frac{(2m+n-1)!}{(n-1)!}[y^{2m}]\left(\frac{y}{\sin y}\right)^{2m+n}.$$

Then, the last coefficient can be determined by

$$[y^{2m}]\left(\frac{y}{\sin y}\right)^{2m+n} = [y^{2m}]\left\{1-\left(1-\frac{\sin y}{y}\right)\right\}^{-2m-n}$$

$$= \sum_{j=0}^{m}(-1)^j\binom{-2m-n}{j}[y^{2m}]\left(1-\frac{\sin y}{y}\right)^j$$

$$= \sum_{j=0}^{m}(-1)^j\binom{-2m-n}{j}\sum_{k=0}^{j}(-1)^k\binom{j}{k}[y^{2m}]\left(\frac{\sin y}{y}\right)^k$$

$$= \sum_{k=0}^{m}\binom{-2m-n}{k}[y^{2m+k}]\sin^k y\sum_{j=k}^{m}(-1)^{j-k}\binom{-2m-n-k}{j-k}$$

$$= \sum_{k=0}^{m}\frac{k!}{(2m+k)!}\binom{-2m-n}{k}\binom{3m+n}{m-k} V(2m+k,k).$$

By simplifying the binomial product

$$\frac{k!(2m+n-1)!}{(n-1)!(2m+k)!}\binom{-2m-n}{k}\binom{3m+n}{m-k} = \binom{3m+n}{m-k}\binom{-n}{2m+k},$$

we confirm the first alternative expression displayed in (52).

Substituting (50) into (52), we derive the explicit formula below

$$\mathcal{V}(2m+n,n) = \sum_{k=0}^{m}\binom{3m+n}{m-k}\binom{-n}{2m+k}\sum_{j=0}^{k}(-1)^{m+j}\binom{k}{j}\frac{(k-2j)^{2m+k}}{2^k \cdot k!},$$

(53)

which is equivalent to the one obtained by Chu and Marini [35]. In addition, it is not hard to verify that the above double sum has the same value as the next one

$$\mathcal{V}(2m+n,n) = \frac{(3m+n)!}{m!(n-1)!}\sum_{k=0}^{m}\sum_{j=0}^{k}\frac{(-1)^{m+k+j}}{2m+n+k}\binom{m}{k}\binom{k}{j}\frac{(k-2j)^{2m+k}}{2^k(2m+k)!}.$$

(54)

### 5.3. Multiple Zeta Series and Asymptotics

The above expression simplifies significantly the following multiple sum representations derived by Borwein and Chamberland [36] (see also Muzaffar [37])

$$\frac{(\arcsin y)^{2n+1}}{(2n+1)!} = \sum_{m=n}^{\infty} \frac{y^{2m+1}\binom{2m}{m}}{4^{m+n}(2m+1)} \sum_{1\le m_1<m_2<\cdots<m_n\le m} \prod_{i=1}^{n} \frac{1}{(m_i-\frac{1}{2})^2}, \quad [G_n(m)];$$

$$\frac{(\arcsin y)^{2n+2}}{(2n+2)!} = \sum_{m=n}^{\infty} \frac{4^{m-n}y^{2m+2}}{\binom{2m+2}{m+1}(m+1)^2} \sum_{1\le m_1<m_2<\cdots<m_n\le m} \prod_{i=1}^{n} \frac{1}{m_i^2}, \quad [H_n(m)];$$

which can be derived by making use of the hypergeometric series (cf. [38]):

$$_2F_1\left[\begin{matrix} \frac{x}{2}, & -\frac{x}{2} \\ & \frac{1}{2} \end{matrix} \,\middle|\, y^2\right] = \cos(x\arcsin y),$$

$$_2F_1\left[\begin{matrix} \frac{1+x}{2}, & \frac{1-x}{2} \\ & \frac{3}{2} \end{matrix} \,\middle|\, y^2\right] = \frac{\sin(x\arcsin y)}{xy}.$$

By comparing these power series, we find two evaluations of multiple sums:

$$\sum_{1\le m_1<m_2<\cdots<m_n\le m} \prod_{i=1}^{n} \frac{1}{(m_i-\frac{1}{2})^2} = \frac{4^{m+n}m!^2}{(2m)!^2}\mathcal{V}(2m+1,2n+1),$$

$$\sum_{1\le m_1<m_2<\cdots<m_n\le m} \prod_{i=1}^{n} \frac{1}{m_i^2} = \frac{4^{n-m}}{m!^2}\mathcal{V}(2m+2,2n+2).$$

The two exceptional cases corresponding to $n=0$

$$\mathcal{V}(2m+1,1) = \frac{(2m)!^2}{4^m m!^2} \quad \text{and} \quad \mathcal{V}(2m+2,2) = 4^m m!^2$$

yield two novel interesting binomial identities:

$$\sum_{i=0}^{m}\sum_{j=0}^{i} \frac{(-1)^{m+i+j}}{2m+i+1}\binom{m}{i}\binom{i}{j}\frac{(i-2j)^{2m+i}}{2^i(2m+i)!} = \frac{4^m m!(\frac{1}{2})_m^2}{(3m+1)!}, \tag{55}$$

$$\sum_{i=0}^{m}\sum_{j=0}^{i} \frac{(-1)^{m+i+j}}{2m+i+2}\binom{m}{i}\binom{i}{j}\frac{(i-2j)^{2m+i}}{2^i(2m+i)!} = \frac{4^m m!^3}{(3m+2)!}. \tag{56}$$

Recall the infinite products

$$\frac{\sin \pi x}{\pi x} = \prod_{n=1}^{\infty}\left\{1-\frac{x^2}{n^2}\right\} \quad \text{and} \quad \cos \pi x = \prod_{n=1}^{\infty}\left\{1-\frac{x^2}{(n-\frac{1}{2})^2}\right\}.$$

By extracting the coefficient of $x^{2n}$, we recover the two infinite series identities (now called multifold Euler sums, see [39,40]), where the first one is due to Zagier [41]:

$$\sum_{1\le m_1<m_2<\cdots<m_n<\infty} \prod_{i=1}^{n} \frac{1}{m_i^2} = \frac{\pi^{2n}}{(2n+1)!},$$

$$\sum_{1\le m_1<m_2<\cdots<m_n<\infty} \prod_{i=1}^{n} \frac{1}{(m_i-\frac{1}{2})^2} = \frac{\pi^{2n}}{(2n)!}.$$

From these multiple zeta evaluations, we find, as $m \to \infty$, two asymptotic formulae:

$$\mathcal{V}(2m+1, 2n+1) \approx \frac{(2m)!^2}{4^{m+n}m!^2} \frac{\pi^{2n}}{(2n)!},$$

$$\mathcal{V}(2m+2, 2n+2) \approx \frac{m!^2}{4^{n-m}} \frac{\pi^{2n}}{(2n+1)!}.$$

By putting them in conjunction with (54), we find, after the replacement $m \to m+n$, the following two equivalent limiting relations:

$$\frac{\pi^{2n}}{2^{2n}} = \lim_{m\to\infty} \frac{(1+3m+2n)!}{4^{m+n}m!(\frac{1}{2})^2_{m+n}}$$
$$\times \sum_{i=0}^{m} \sum_{j=0}^{i} \frac{(-1)^{i+j+m}}{1+2m+2n+i} \binom{m}{i} \binom{i}{j} \frac{(i-2j)^{2m+i}}{2^i(2m+i)!},$$

$$\frac{\pi^{2n}}{2^{2n}} = \lim_{m\to\infty} \frac{(2+3m+2n)!}{4^{m+n}m!(m+n)!^2}$$
$$\times \sum_{i=0}^{m} \sum_{j=0}^{i} \frac{(-1)^{i+j+m}}{2+2m+2n+i} \binom{m}{i} \binom{i}{j} \frac{(i-2j)^{2m+i}}{2^i(2m+i)!}.$$

An intriguing question is how to prove them directly and independently.

## 6. arctan and Multiple Zeta Values

We shall examine, in this section, the reciprocal function pair tan and arctan. Define the connection coefficients by exponential generating functions as follows

$$\frac{(\tan x)^n}{n!} = \sum_{m=0}^{\infty} \frac{x^{2m+n}}{(2m+n)!} \mathrm{T}(2m+n, n),$$
$$\frac{(\arctan x)^n}{n!} = \sum_{m=0}^{\infty} \frac{x^{2m+n}}{(2m+n)!} \mathcal{T}(2m+n, n);$$

(57)

which can be expressed as

$$\mathrm{T}(2m+n, n) = \frac{(2m+n)!}{n!} [x^{2m+n}](\tan x)^n,$$
$$\mathcal{T}(2m+n, n) = \frac{(2m+n)!}{n!} [x^{2m+n}](\arctan x)^n.$$

(58)

It is routine to check that for all the $m, n \in \mathbb{N}$, the following is true

$$\mathcal{T}(2m+n-1, n) = \mathrm{T}(2m+n-1, n) = 0.$$

and

$$\mathrm{T}(2m+n, n), \ (-1)^m \mathcal{T}(2m+n, n) \in \mathbb{N}.$$

### 6.1. Explicit Formulae of $\mathrm{T}(m, n)$ and $\mathcal{T}(m, n)$

The explicit formulae of $\mathrm{T}(m, n)$ and $\mathcal{T}(m, n)$ can be stated as

$$\mathrm{T}(2m+n, n) = \frac{(-4)^m}{n!} \sum_{k=0}^{2m} \frac{(n+k)!}{2^k} \binom{-n}{k} \mathrm{S}_2(2m+n, k+n),$$

$$\mathcal{T}(2m+n, n) = (-1)^m \sum_{k=0}^{2m} (-2)^k \frac{(2m+n)!}{(n+k)!} \binom{-n-k}{2m-k} \mathrm{S}_1(k+n, n).$$

(59)

These two formulae are equivalent to those for the hyperbolic tanh and arctanh that appear in [42]. For different properties and recurrence relations, the reader can consult Comtet [2] (Chapter 6: Exercise 11), Cvijovic [43], Hardtke [44], and Lomont [45].

**Proof.** In terms of the exponential function, we have

$$
(\tan x)^n = (\sqrt{-1})^n \left\{ \frac{1 - e^{2x\sqrt{-1}}}{1 + e^{2x\sqrt{-1}}} \right\}^n
$$

$$
= (\sqrt{-1})^n \left\{ \frac{(1 - e^{2x\sqrt{-1}})/2}{1 - (1 - e^{2x\sqrt{-1}})/2} \right\}^n
$$

$$
= (\sqrt{-1})^n \sum_{k=0}^{\infty} (-1)^k \binom{-n}{k} \frac{(1 - e^{2x\sqrt{-1}})^{n+k}}{2^{n+k}}.
$$

Then, we can rewrite the coefficient as

$$
T(2m + n, n) = \frac{(2m+n)!}{n!} [x^{2m+n}] (\tan x)^n
$$

$$
= \frac{(2m+n)!}{n!} (\sqrt{-1})^n \sum_{k=0}^{2m} (-1)^k \binom{-n}{k} [x^{2m+n}] \frac{(1 - e^{2x\sqrt{-1}})^{n+k}}{2^{n+k}}.
$$

Observing further that

$$
[x^{2m+n}] \frac{(e^{2x\sqrt{-1}} - 1)^{n+k}}{2^{n+k}} = \frac{(2\sqrt{-1})^{2m+n}}{2^{n+k}} \frac{(k+n)!}{(2m+n)!} S_2(2m+n, k+n),
$$

we derive the first formula in terms of the Stirling numbers of the second kind

$$
T(2m + n, n) = \frac{(-4)^m}{n!} \sum_{k=0}^{2m} \frac{(n+k)!}{2^k} \binom{-n}{k} S_2(2m+n, k+n). \tag{60}
$$

At the same time, according to the expansion

$$
(\arctan x)^n = \left( \frac{\sqrt{-1}}{2} \right)^n \ln^n \left( 1 - \frac{2x\sqrt{-1}}{1 + x\sqrt{-1}} \right)
$$

$$
= \left( \frac{\sqrt{-1}}{2} \right)^n \sum_{k=0}^{\infty} \frac{n!}{(n+k)!} \left( \frac{-2x\sqrt{-1}}{1 + x\sqrt{-1}} \right)^{n+k} S_1(k+n, n),
$$

the coefficient $\mathcal{T}(2m + n, n)$ can be computed as follows:

$$
\mathcal{T}(2m + n, n) = \frac{(2m+n)!}{n!} [x^{2m+n}] (\arctan x)^n
$$

$$
= \left( \frac{\sqrt{-1}}{2} \right)^n \sum_{k=0}^{2m} \frac{(2m+n)!}{(n+k)!} S_1(k+n, n) [x^{2m+n}] \left( \frac{-2x\sqrt{-1}}{1 + x\sqrt{-1}} \right)^{n+k}
$$

$$
= (-1)^m \sum_{k=0}^{2m} (-2)^k \frac{(2m+n)!}{(n+k)!} \binom{-n-k}{2m-k} S_1(k+n, n).
$$

This completes the proof of two formulae displayed in (59). □

### 6.2. Multiple Harmonic Numbers $H_m^{\langle n \rangle}$

Milgram [46] found and then Chen [47] rederived the following expansion in terms of multiple harmonic numbers

$$\frac{\arctan^n x}{n!} = \sum_{m=0}^{\infty} (-1)^m \frac{x^{2m+n}}{2m+n} H_m^{\langle n-1 \rangle} = \sum_{0 \le m_1 \le m_2 \le \cdots \le m_n < \infty} \frac{(-1)^{m_n} x^{2m_n+n}}{\prod_{j=1}^{n} (2m_j + j)}, \tag{61}$$

where the multiple harmonic number is defined by

$$H_m^{\langle n \rangle} = \sum_{0 \le m_1 \le m_2 \le \cdots \le m_n \le m} \prod_{j=1}^{n} \frac{1}{2m_j + j}.$$

Here, we offer a simpler proof by induction. When $n = 1$, the result is true because

$$\arctan x = \sum_{m=0}^{\infty} (-1)^m \frac{x^{2m+1}}{2m+1}$$

which also follows from that

$$\frac{1}{1+x^2} = D_x \arctan x = \sum_{m=0}^{\infty} (-1)^m x^{2m}.$$

For $n = 2$, it can be checked as follows. Observing that

$$\frac{\arctan^2 x}{2} = \int_0^x D_x \frac{\arctan^2 x}{2} dx = \int_0^x \frac{\arctan x}{1+x^2} dx$$

we have

$$\frac{\arctan x}{1+x^2} = \sum_{k=0}^{\infty} (-1)^k \frac{x^{2k+1}}{2k+1} \sum_{i=0}^{\infty} (-1)^i x^{2i}$$

$$= \sum_{m=0}^{\infty} (-1)^m x^{2m+1} \sum_{k=0}^{m} \frac{1}{2k+1}$$

$$= \sum_{m=0}^{\infty} (-1)^m x^{2m+1} H_m^{\langle 1 \rangle}.$$

This leads us to the expression

$$\frac{\arctan^2 x}{2} = \int_0^x \frac{\arctan x}{1+x^2} dx = \sum_{m=0}^{\infty} (-1)^m \frac{x^{2m+2}}{2m+2} H_m^{\langle 1 \rangle}.$$

Supposing the Formula (61) is valid for $n-1$, we can analogously proceed for $n$ in the following manner:

$$\frac{\arctan^n x}{n!} = \int_0^x D_x \frac{\arctan^n x}{n!} dx = \int_0^x \frac{\arctan^{n-1} x}{(n-1)!(1+x^2)} dx$$

$$= \int_0^x \sum_{k=0}^{\infty} (-1)^k \frac{x^{2k+n}}{2k+n} H_k^{\langle n-1 \rangle} \sum_{i=0}^{\infty} (-1)^i x^{2i}$$

$$= \int_0^x \sum_{m=0}^{\infty} (-1)^m x^{2m+n} \sum_{k=0}^{m} \frac{H_k^{\langle n-1 \rangle}}{2k+n}$$

$$= \sum_{m=0}^{\infty} (-1)^m \frac{x^{2m+n+1}}{2m+n+1} \sum_{k=0}^{m} \frac{H_k^{\langle n-1 \rangle}}{2k+n}.$$

Consequently, (61) follows from the recurrence relation below:

$$\sum_{k=0}^{m} \frac{H_k^{\langle n-1\rangle}}{2k+n} = H_m^{\langle n\rangle}.$$

Furthermore, by comparing (59) with (61), we obtain the identity

$$H_m^{\langle n-1\rangle} = (-1)^m \frac{T(2m+n,n)}{(2m+n-1)!}$$

$$= \sum_{k=0}^{2m} (-2)^k \frac{2m+n}{(n+k)!} \binom{-n-k}{2m-k} S_1(k+n,n).$$

The first two cases for $n = 1$ and $n = 2$ result in two binomial sums

$$\sum_{k=0}^{2m} (-2)^k \frac{2m+1}{(k+1)!} \binom{-k-1}{2m-k} S_1(k+1,1) = 1,$$

$$\sum_{k=0}^{2m} (-2)^k \frac{2m+2}{(k+2)!} \binom{-k-2}{2m-k} S_1(k+2,2) = H_{2m+1} - \frac{H_m}{2};$$

where $H_m$ is the usual harmonic number. Their proofs are not difficult because

$$S_1(k+1,1) = k!(-1)^k \quad \text{and} \quad S_1(k+2,2) = (-1)^k(k+1)!H_{k+1}.$$

## 7. Composite Series Pairs

As we mentioned previously that if $\{f,g\}$ and $(F,G)$ are two reciprocal pairs, then both $\{-f(-x), -g(-x)\}$ and $\{F(f), g(G)\}$ are again reciprocal pairs. Some pairs of compositional series will be considered now.

### 7.1. **tan** *and* **arctan**

Consider the inverse pair given by

$$F(x) = \frac{x}{\sqrt{1-x^2}} \quad \text{and} \quad G(x) = \frac{x}{\sqrt{1+x^2}}.$$

Their connection coefficients are determined by

$$\left(\frac{x}{\sqrt{1-x^2}}\right)^n = \sum_{m=0}^{\infty} (-1)^m \binom{-\frac{n}{2}}{m} x^{2m+n},$$

$$\left(\frac{x}{\sqrt{1+x^2}}\right)^n = \sum_{m=0}^{\infty} \binom{-\frac{n}{2}}{m} x^{2m+n}.$$

(62)

Because $f(x) = \sin x$ and $g(x) = \arcsin x$ are another inverse pair, we have the following compositional inverse pair:

$$\tan x = F(f(x)) \quad \text{and} \quad \arctan x = g(G(x)).$$

The connection coefficients can be expressed as

$$
\begin{aligned}
\mathrm{T}(2m+n,n) &= \frac{\lfloor \tan^n x; 2m+n \rceil}{n!} \\
&= \sum_{k=0}^{m} \left\lfloor \left(\frac{x}{\sqrt{1-x^2}}\right)^n; 2k+n \right\rceil \frac{\lfloor \sin^{2k+n} x; 2m+n \rceil}{n!(2k+n)!} \\
&= \sum_{k=0}^{m} (-1)^k \frac{(2k+n)!}{n!} \binom{-\frac{n}{2}}{k} \mathrm{V}(2m+n, 2k+n); \\
\mathcal{T}(2m+n,n) &= \frac{\lfloor \arctan^n x; 2m+n \rceil}{n!} \\
&= \sum_{k=0}^{m} \frac{\lfloor \arcsin^n x; 2k+n \rceil}{n!(2k+n)!} \left\lfloor \left(\frac{x}{\sqrt{1+x^2}}\right)^{2k+n}; 2m+n \right\rceil \\
&= \sum_{k=0}^{m} \frac{(2m+n)!}{(2k+n)!} \binom{-k-\frac{n}{2}}{m-k} \mathcal{V}(2k+n, n).
\end{aligned}
$$

Then, we have the following expressions of double and triple sums:

$$
\mathrm{T}(2m+n,n) = \sum_{k=0}^{m} \sum_{i=0}^{n+2k} (-1)^{m-i} \binom{-\frac{n}{2}}{k} \binom{n+2k}{i} \frac{(n+2k-2i)^{2m+n}}{2^{n+2k} n!}, \tag{63}
$$

$$
\mathcal{T}(2m+n,n) = \frac{(2m+n)!}{(n-1)!} \sum_{k=0}^{m} \sum_{i=0}^{k} \sum_{j=0}^{i} \frac{(-1)^{k+i+j}}{2k+n+i} \binom{-k-\frac{n}{2}}{m-k} \tag{64}
$$

$$
\times \binom{3k+n}{k} \binom{k}{i} \binom{i}{j} \frac{(i-2j)^{2k+i}}{2^i (2k+i)!}.
$$

### 7.2. Bernoulli Numbers of Higher Orders

Bernoulli numbers of higher orders are defined by

$$
\left(\frac{x}{e^x-1}\right)^n = \sum_{m=0}^{\infty} \frac{x^m}{m!} \mathrm{B}_m^n \quad \text{and} \quad \frac{x}{2}\coth\frac{x}{2} = \sum_{m=0}^{\infty} \frac{x^{2m}}{(2m)!} \mathrm{B}_{2m}. \tag{65}
$$

There exist numerous research papers around these numbers. The interested reader can refer, for example, to [42,48–53].

Let $\varphi(x) = \frac{e^x-1}{x}$. It is crucial to check that the series $f(x) = x/\varphi(x)$ determines implicitly its inverse series $g(x)$. Then, $\{f(x), g(x)\}$ form a reciprocal pair with the following connection coefficients:

$$
\mathrm{B}_m^n = m! \langle f^n; m+n \rangle \quad \text{and} \quad \mathcal{B}_m^n = m! \langle g^n; m+n \rangle.
$$

These coefficients satisfy $\mathcal{B}_m^n > 0$ and $\mathcal{B}_m^n \in \mathbb{Q}$ and admit the explicit formulae

$$
\mathrm{B}_m^n = \sum_{k=0}^{m} \frac{n! k!}{(n+k)!} \mathrm{S}_1(n+k,n) \mathrm{S}_2(m,k), \tag{66}
$$

$$
\mathcal{B}_m^n = \frac{n}{m+n} \frac{m!(m+n)!}{(2m+n)!} \mathrm{S}_2(2m+n, m+n). \tag{67}
$$

The first one extends the Worpitzky formula [54] (1883) for Bernoulli numbers

$$
\mathrm{B}_m = \sum_{k=0}^{m} (-1)^k \frac{k!}{k+1} \mathrm{S}_2(m,k).
$$

**Proof.** Observing the composite function

$$\frac{x}{e^x - 1} = \frac{\ln(1+y)}{y} \quad \text{and} \quad y = e^x - 1$$

we can manipulate the series

$$\begin{aligned}
\mathrm{B}_m^n = m! \langle f^n; m+n \rangle &= m! \left\langle \frac{x^n}{(e^x - 1)^n}; m \right\rangle \\
&= m! \sum_{k=0}^m \left\langle \frac{\ln^n(1+x)}{x^n}; k \right\rangle \left\langle (e^x - 1)^k; m \right\rangle \\
&= \sum_{k=0}^m \frac{n! k!}{(n+k)!} \mathrm{S}_1(n+k, n) \mathrm{S}_2(m, k),
\end{aligned}$$

which gives rise to the first formula. Instead, the "dual Bernoulli numbers of order $n$" are easier to handle by taking into account the Lagrange expansion Formula (1)

$$\begin{aligned}
\mathcal{B}_m^n = m! \langle g^n; m+n \rangle &= \frac{m! n}{m+n} \left\langle \frac{(e^x - 1)^{m+n}}{x^{m+n}}; m \right\rangle \\
&= \frac{n}{m+n} \frac{m!(m+n)!}{(2m+n)!} \mathrm{S}_2(2m+n, m+n),
\end{aligned}$$

and the exponential generating function of Stirling numbers of the second kind.  □

These numbers satisfy the recurrence relations (13)

$$\mathrm{B}_m^n = \sum_{k=0}^m \binom{n}{k} \binom{m-n}{m-k} \mathrm{B}_m^k \quad \text{and} \quad \mathcal{B}_m^n = \sum_{k=0}^m \binom{n}{k} \binom{m-n}{m-k} \mathcal{B}_m^k$$

as well as the reciprocal relations (21)

$$\mathrm{B}_m^n = \frac{n}{m+n} \sum_{k=0}^m \binom{-m-n}{k} \binom{2m+n}{m-k} \mathcal{B}_m^k, \tag{68}$$

$$\mathcal{B}_m^n = \frac{n}{m+n} \sum_{k=0}^m \binom{-m-n}{k} \binom{2m+n}{m-k} \mathrm{B}_m^k. \tag{69}$$

*7.3. Euler Numbers of Higher Orders*

Euler numbers of higher orders are defined by

$$\left( \frac{2e^x}{e^{2x} + 1} \right)^n = \sum_{m=0}^\infty \frac{x^m}{m!} \mathrm{E}_m^n, \tag{70}$$

where $\mathrm{E}_m^n = 0$ for odd $m \in \mathbb{N}$, since $\operatorname{sech} x$ is an even function. More information about these numbers can be found in [53,55–59].

Let $\varphi(x) = \cosh x$. It is trivial to check that the series $f(x) = x/\varphi(x)$ determines implicitly its inverse series $g(x)$. Then, the reciprocal pair $\{f(x), g(x)\}$ gives rise to the following connection coefficients

$$\mathrm{E}_m^n = m! \langle f^n; m+n \rangle \quad \text{and} \quad \mathcal{E}_m^n = m! \langle g^n; m+n \rangle$$

with $(-1)^m \mathrm{E}_{2m}^n \in \mathbb{N}$ and $\mathcal{E}_{2m}^n \in \mathbb{N}$, as well as the explicit formulae

$$\mathrm{E}_{2m}^n = \sum_{k=0}^{m} 2^{1+2m-2k} \binom{\frac{-n}{2}}{k} \sum_{i=0}^{k} (-1)^i \binom{2k}{i} (k-i)^{2m}, \tag{71}$$

$$\mathcal{E}_{2m}^n = \frac{n}{2m+n} \sum_{k=0}^{2m+n} \binom{2m+n}{k} \frac{(2m+n-2k)^{2m}}{2^{2m+n}}. \tag{72}$$

**Proof.** Observing the composite function

$$\operatorname{sech} x = \frac{1}{\sqrt{1+y^2}} \quad \text{and} \quad y = \sinh x$$

we can manipulate the series

$$\begin{aligned}
\mathrm{E}_{2m}^n &= (2m)! \langle f^n; 2m+n \rangle = (2m)! \langle \operatorname{sech}^n x; 2m \rangle \\
&= (2m)! \sum_{k=0}^{m} \left\langle (1+x^2)^{-n/2}; 2k \right\rangle \left\langle \sinh^{2k} x; 2m \right\rangle \\
&= \sum_{k=0}^{m} (-1)^{m-k} \binom{\frac{-n}{2}}{k} (2k)! \mathrm{V}(2m, 2k),
\end{aligned}$$

where we have employed, according to (50), the fact that

$$\langle \sinh^n x; 2m+n \rangle = \frac{n!(-1)^m}{(2m+n)!} \mathrm{V}(2m+n, n).$$

This proves Formula (71). The "dual Euler numbers of order $n$" is evaluated by making use of the Lagrange inversion Formula (1):

$$\begin{aligned}
\mathcal{E}_{2m}^n &= (2m)! \langle g^n; 2m+n \rangle = \frac{(2m)!n}{2m+n} \left\langle \cosh^{2m+n} x; 2m \right\rangle \\
&= \frac{(2m)!n}{2m+n} \sum_{k=0}^{2m+n} \binom{2m+n}{k} \frac{(2m+n-2k)^{2m}}{2^{2m+n}(2m)!}.
\end{aligned}$$

The ultimate coefficient has been evaluated as follows:

$$\begin{aligned}
\left\langle \cosh^{2m+n} x; 2m \right\rangle &= [x^{2m}] \frac{(e^x + e^{-x})^{2m+n}}{2^{2m+n}} \\
&= \sum_{k=0}^{2m+n} \binom{2m+n}{k} \frac{[x^{2m}]}{2^{2m+n}} e^{x(2m+n-2k)} \\
&= \sum_{k=0}^{2m+n} \binom{2m+n}{k} \frac{(2m+n-2k)^{2m}}{2^{2m+n}(2m)!},
\end{aligned}$$

which is justified by the binomial expansion and the Maclaurin series. □

These numbers satisfy the recurrence relations (13)

$$\mathrm{E}_{2m}^n = \sum_{k=0}^{2m} \binom{n}{k} \binom{2m-n}{2m-k} \mathrm{E}_{2m}^k \quad \text{and} \quad \mathcal{E}_{2m}^n = \sum_{k=0}^{2m} \binom{n}{k} \binom{2m-n}{2m-k} \mathcal{E}_{2m}^k$$

as well as the reciprocal relations (21)

$$E_{2m}^n = \frac{n}{2m+n} \sum_{k=0}^{2m} \binom{-2m-n}{k} \binom{4m+n}{2m-k} \mathcal{E}_{2m}^k,$$ (73)

$$\mathcal{E}_{2m}^n = \frac{n}{2m+n} \sum_{k=0}^{2m} \binom{-2m-n}{k} \binom{4m+n}{2m-k} E_{2m}^k.$$ (74)

*7.4. Cauchy Numbers of the First Kind*

The Cauchy numbers of the first kind (see Comtet [2] (p. 294) and [60,61]) are defined by the integral and the generating function:

$$C_n = \int_0^1 \langle x \rangle_n dx \quad \text{with} \quad \frac{x}{\ln(1+x)} = \sum_{n=0}^{\infty} \frac{x^n}{n!} C_n.$$

Their higher-order counterparts (see Zhao [62]) are given by

$$\left( \frac{x}{\ln(1+x)} \right)^n = \sum_{m=0}^{\infty} \frac{x^m}{m!} C_m^n.$$ (75)

Let $\varphi(x) = \frac{\ln(1+x)}{x}$. It is trivial to check that the series $f(x) = x/\varphi(x)$ determines implicitly its inverse series $g(x)$. Then, the reciprocal pair $\{f(x), g(x)\}$ produces the following connection coefficients

$$C_m^n = m! \langle f^n; m+n \rangle \quad \text{and} \quad \mathcal{C}_m^n = m! \langle g^n; m+n \rangle$$

with $(-1)^m \mathcal{C}_m^n > 0$ and $(-1)^m \mathcal{C}_m^n \in \mathbb{Q}$, as well as the explicit formulae

$$C_m^n = \sum_{k=0}^{m} \frac{n! k!}{(n+k)!} S_1(m,k) S_2(n+k,n),$$ (76)

$$\mathcal{C}_m^n = \frac{n}{m+n} \frac{m!(m+n)!}{(2m+n)!} S_1(2m+n, m+n).$$ (77)

These numbers resemble the Bernoulli numbers of higher orders in the sense that the Stirling numbers of first and second kind exchange their roles.

**Proof.** Observing the composite function

$$\frac{x}{\ln(1+x)} = \frac{e^y - 1}{y} \quad \text{and} \quad y = \ln(1+x),$$

we can manipulate the series

$$
\begin{aligned}
C_m^n &= m! \langle f^n; m+n \rangle = m! \left\langle \frac{x^n}{\ln^n(1+x)}; m \right\rangle \\
&= m! \sum_{k=0}^{m} \left\langle \frac{(e^x-1)^n}{x^n}; k \right\rangle \left\langle \ln^k(1+x); m \right\rangle \\
&= \sum_{k=0}^{m} \frac{n! k!}{(n+k)!} S_1(m,k) S_2(n+k,n),
\end{aligned}
$$

which gives the first formula. The "dual Cauchy numbers of the first kind of order $n$" are obtained by making use of the Lagrange inversion Formula (1)

$$\mathcal{C}_m^n = m!\langle g^n; m+n\rangle = \frac{m!n}{m+n}\left\langle \frac{\ln^{m+n}(1+x)}{x^{m+n}}; m\right\rangle$$

$$= \frac{n}{m+n}\frac{m!(m+n)!}{(2m+n)!}S_1(2m+n, m+n),$$

and the exponential generating function of Stirling numbers of the first kind.  □

These numbers satisfy the recurrence relations (13)

$$C_m^n = \sum_{k=0}^{m}\binom{n}{k}\binom{m-n}{m-k}C_m^k \quad\text{and}\quad \mathcal{C}_m^n = \sum_{k=0}^{m}\binom{n}{k}\binom{m-n}{m-k}\mathcal{C}_m^k$$

as well as the reciprocal relations (21)

$$C_m^n = \frac{n}{m+n}\sum_{k=0}^{m}\binom{-m-n}{k}\binom{2m+n}{m-k}\mathcal{C}_m^k, \tag{78}$$

$$\mathcal{C}_m^n = \frac{n}{m+n}\sum_{k=0}^{m}\binom{-m-n}{k}\binom{2m+n}{m-k}C_m^k. \tag{79}$$

### 7.5. Cauchy Numbers of the Second Kind

The Cauchy numbers of the second kind (see Comtet [2] (p. 294) and [61,63]) are defined by the integral and the generating function:

$$K_n = \int_0^1 (x)_n dx \quad\text{with}\quad \frac{x}{(x-1)\ln(1-x)} = \sum_{n=0}^{\infty}\frac{x^n}{n!}K_n.$$

Their higher-order counterparts can be analogously given by

$$\left(\frac{x}{(x-1)\ln(1-x)}\right)^n = \sum_{m=0}^{\infty}\frac{x^m}{m!}K_m^n. \tag{80}$$

Let $\varphi(x) = \frac{(x-1)\ln(1-x)}{x}$. It is trivial to check that the series $f(x) = x/\varphi(x)$ determines implicitly its inverse series $g(x)$. Then, $\{f(x), g(x)\}$ form a reciprocal pair with the following connection coefficients:

$$K_m^n = m!\langle f^n; m+n\rangle \quad\text{and}\quad \mathcal{K}_m^n = m!\langle g^n; m+n\rangle.$$

For these numbers, we have also $\mathcal{K}_m^n > 0$ and $\mathcal{K}_m^n \in \mathbb{Q}$, as well as the explicit formulae

$$K_m^n = \sum_{k=0}^{m}(-1)^{m-k}\frac{n!k!}{(n+k)!}S_1(m, k)S_2(n+k, n), \tag{81}$$

$$\mathcal{K}_m^n = (-1)^m\frac{m!n}{m+n}\sum_{k=n}^{m+n}\frac{(m+n)!}{(m+k)!}\binom{m+n}{k}S_1(m+k, m+n). \tag{82}$$

It is curious that the explicit formulae for the higher-order Cauchy numbers of the two kinds differ only in the alternating sign in their summands.

**Proof.** Observing the composite function

$$\frac{x}{(x-1)\ln(1-x)} = \frac{e^y - 1}{y} \quad\text{and}\quad y = -\ln(1-x),$$

we can manipulate the series

$$
\begin{aligned}
\mathrm{K}_m^n = m! \langle f^n; m+n \rangle &= m! \left\langle \frac{x^n}{(x-1)^n \ln^n(1-x)}; m \right\rangle \\
&= m! \sum_{k=0}^m (-1)^k \left\langle \frac{(e^x-1)^n}{x^n}; k \right\rangle \left\langle \ln^k(1-x); m \right\rangle \\
&= \sum_{k=0}^m (-1)^{m-k} \frac{n!k!}{(n+k)!} \mathrm{S}_1(m,k) \mathrm{S}_2(n+k,n),
\end{aligned}
$$

which gives the first formula. The "dual Cauchy numbers of the second kind of order $n$" is derived by making use of the Lagrange expansion formula

$$
\begin{aligned}
\mathcal{K}_m^n = m! \langle g^n; m+n \rangle &= \frac{m!n}{m+n} \left\langle \frac{(x-1)^{m+n} \ln^{m+n}(1-x)}{x^{m+n}}; m \right\rangle \\
&= \frac{m!n}{m+n} \left\langle (x-1)^{m+n} \ln^{m+n}(1-x); 2m+n \right\rangle \\
&= (-1)^m \frac{m!n}{m+n} \sum_{k=n}^{m+n} \frac{(m+n)!}{(m+k)!} \binom{m+n}{k} \mathrm{S}_1(m+k, m+n),
\end{aligned}
$$

and the generating function of Stirling numbers of the second kind.  $\square$

According to (13), these numbers satisfy the recurrence relations

$$
\mathrm{K}_m^n = \sum_{k=0}^m \binom{n}{k} \binom{m-n}{m-k} \mathrm{K}_m^k \quad \text{and} \quad \mathcal{K}_m^n = \sum_{k=0}^m \binom{n}{k} \binom{m-n}{m-k} \mathcal{K}_m^k
$$

as well as the reciprocal relations (21)

$$
\mathrm{K}_m^n = \frac{n}{m+n} \sum_{k=0}^m \binom{-m-n}{k} \binom{2m+n}{m-k} \mathcal{K}_m^k, \tag{83}
$$

$$
\mathcal{K}_m^n = \frac{n}{m+n} \sum_{k=0}^m \binom{-m-n}{k} \binom{2m+n}{m-k} \mathrm{K}_m^k. \tag{84}
$$

## 8. Further Exploration

In mathematics, a Sheffer sequence (or poweroid, see [64] §4.3) is a polynomial sequence $\{P_n(x)\}_{n \geq 0}$ in which the index of each polynomial equals its degree and characterized by its exponential generating function

$$
\phi(y) e^{x\lambda(y)} = \sum_{n=0}^{\infty} \frac{y^n}{n!} P_n(x),
$$

where $\phi(y)$ and $\lambda(y)$ are power series in $y$.

Examples of polynomial sequences which are Sheffer sequences (see Roman [64]):

- The monomials, i.e., $\{x^n : n = 0, 1, 2, \cdots\}$:

$$
e^{xy} = \sum_{n=0}^{\infty} \frac{y^n}{n!} x^n.
$$

- The Abel polynomials, i.e., $P_n(x) = x(x-\beta n)^{n-1}$:

$$
e^{x\tau} = \sum_{n=0}^{\infty} x(x-\beta n)^{n-1} \frac{y^n}{n!} \quad \text{with} \quad y = \tau e^{\beta \tau}.
$$

- The Gould polynomials, i.e., $P_n(x) = \frac{x}{x-\beta n}\binom{x-\beta n}{n}$

$$(1+\tau)^x = \sum_{n=0}^{\infty} \frac{x}{x-\beta n}\binom{x-\beta n}{n}y^n \quad \text{with} \quad y = \tau(1+\tau)^\beta.$$

- The Bernoulli polynomials (cf. [52]):

$$\frac{ye^{xy}}{e^y-1} = \sum_{n=0}^{\infty} \frac{y^n}{n!}B_n(x).$$

- The Euler polynomials:

$$\frac{2e^{xy}}{e^y+1} = \sum_{n=0}^{\infty} \frac{y^n}{n!}E_n(x).$$

- The central factorial polynomials:

$$e^{2x\,\mathrm{arcsinh}(y/2)} = \sum_{n=0}^{\infty} \frac{y^n}{n!}Z_n(x).$$

- The Hermite polynomials:

$$e^{2xy-y^2} = \sum_{n=0}^{\infty} \frac{y^n}{n!}H_n(x).$$

- The Charlier polynomials:

$$e^y(1-y/a)^x = \sum_{n=0}^{\infty} \frac{y^n}{n!}C_n(x,a).$$

- The Laguerre polynomials:

$$\frac{\exp\left(\frac{xy}{y-1}\right)}{(1-y)^{\alpha+1}} = \sum_{n=0}^{\infty} L_n^\alpha(x)y^n.$$

- The Mahler polynomials:

$$\exp\left(x(1+y-e^y)\right) = \sum_{n=0}^{\infty} \frac{y^n}{n!}G_n(x).$$

- The Mott polynomials:

$$\exp\left(x\frac{\sqrt{1-y^2}-1}{y}\right) = \sum_{n=0}^{\infty} \frac{y^n}{n!}Q_n(x).$$

- The Stirling polynomials (cf. [27,29]):

$$\left(\frac{ye^y}{e^y-1}\right)^{1+x} = \sum_{n=0}^{\infty} \frac{y^n}{n!}S_n(x).$$

- Meixner polynomials:

$$\frac{(1-y/c)^x}{(1-y)^{\beta+x}} = \sum_{n=0}^{\infty} \frac{(\beta)_n y^n}{n!}M_n(x,\beta,c).$$

- Meixner–Pollaczek polynomials:

$$(1 - ye^{\phi\sqrt{-1}})^{-\lambda+x\sqrt{-1}}(1 - ye^{-\phi\sqrt{-1}})^{-\lambda-x\sqrt{-1}} = \sum_{n=0}^{\infty} \frac{y^n}{n!} P_n^{(\lambda)}(x,\phi).$$

By exploring the above generating functions, it is possible to review orthogonality relations, recurrence relations, convolutions, and explicit formulae related to these polynomials. We take the Stirling polynomials as an example to show how to derive, by connection coefficients, the following two interesting explicit formulae:

$$S_n(x) = \binom{x+n}{n} \sum_{k=0}^{n} (-1)^{n-k} \frac{1+x+n}{1+x+k} \frac{n!^2 S_2(n+k,k)}{(n+k)!(n-k)!}$$

$$= \sum_{k=0}^{n} (-1)^{n-k} \frac{\binom{x+k}{k}\binom{x+n+1}{n-k}}{\binom{n+k}{k}} S_2(n+k,k),$$

$$S_n(x) = \binom{x-n}{n} \sum_{k=0}^{n} (-1)^k \frac{1+x}{1+x-n-k} \frac{n!^2 S_1(n+k,k)}{(n+k)!(n-k)!}$$

$$= \frac{1+x}{1+x-n} \sum_{k=0}^{n} (-1)^k \frac{\binom{1+x-n}{k}\binom{x-n-k}{n-k}}{\binom{n+k}{k}} S_1(n+k,k).$$

**Proof.** Observe that

$$S_n(x) = n!\left\langle \left(\frac{1-e^{-y}}{y}\right)^{-1-x}; n\right\rangle = n!\left\langle \left\{1 - \left(1 - \frac{1-e^{-y}}{y}\right)\right\}^{-1-x}; n\right\rangle$$

$$= n!\sum_{i=0}^{n} (-1)^i \binom{-x-1}{i} \left\langle \left(1 - \frac{1-e^{-y}}{y}\right)^i; n\right\rangle$$

$$= n!\sum_{i=0}^{n} (-1)^i \binom{-x-1}{i} \sum_{k=0}^{i} (-1)^k \binom{i}{k} \left\langle \left(\frac{1-e^{-y}}{y}\right)^k; n\right\rangle$$

$$= n!\sum_{i=0}^{n} (-1)^i \binom{-x-1}{i} \sum_{k=0}^{i} (-1)^{n+k} \binom{i}{k} \frac{k!}{(n+k)!} S_2(n+k,k)$$

$$= \sum_{k=0}^{n} (-1)^n \binom{-x-1}{k} \frac{n!k!}{(n+k)!} S_2(n+k,k) \sum_{i=k}^{n} (-1)^{i-k} \binom{-x-k-1}{i-k}.$$

Evaluating the last sum by

$$\sum_{i=k}^{n} (-1)^{i-k} \binom{-x-k-1}{i-k} = (-1)^{n-k} \binom{-x-k-2}{n-k},$$

we obtain an expression equivalent to the first formula.

Now, for the two variables $(\tau, y)$ related by $y = -\ln(1-\tau)$, define

$$y = f(\tau) = \tau/\varphi(\tau) \quad \text{where} \quad \varphi(\tau) = \frac{-\tau}{\ln(1-\tau)} \quad \text{with} \quad \varphi(0) = 1.$$

Then, the inverse function of $y = f(\tau)$ is given by $\tau = g(y) = 1 - e^{-y}$. Define the composite function by

$$F(\tau) = \left\{\frac{\ln(1-\tau)}{-\tau}\right\}^{x+1} = \left\{\frac{y}{1-e^{-y}}\right\}^{x+1}.$$

According to the Lagrange expansion Formula (1), we can extract its coefficient:

$$
\begin{aligned}
S_n(x) &= n! \left\langle \left( \frac{y}{1 - e^{-y}} \right)^{x+1} ; n \right\rangle = n! \left\langle \left( \frac{\ln(1 - \tau)}{-\tau} \right)^{x+1} ; n \right\rangle \\
&= (n-1)! \left\langle F'(\tau) \varphi^n(\tau); n - 1 \right\rangle = (n-1)! \left\langle \varphi^n(\tau) \frac{d}{d\tau} \varphi^{-x-1}(\tau); n - 1 \right\rangle \\
&= \frac{(1+x)(n-1)!}{1 + x - n} \left\langle \frac{d}{d\tau} \varphi^{n-x-1}(\tau); n - 1 \right\rangle = \frac{(1+x)n!}{1 + x - n} \left\langle \varphi^{n-x-1}(\tau); n \right\rangle.
\end{aligned}
$$

For the last coefficient, we can compute it similarly by

$$
\begin{aligned}
\left\langle \varphi^{n-x-1}(\tau); n \right\rangle &= \left\langle \left\{ 1 - \left( 1 - \frac{1}{\varphi(\tau)} \right) \right\}^{1+x-n} ; n \right\rangle \\
&= \sum_{k=0}^{n} (-1)^{n-k} \binom{1+x-n}{k} \binom{x-n-k}{n-k} \left\langle \varphi^{-k}(\tau); n \right\rangle \\
&= \sum_{k=0}^{n} (-1)^{k} \binom{1+x-n}{k} \binom{x-n-k}{n-k} \frac{k!}{(n+k)!} S_1(n+k, k).
\end{aligned}
$$

By substitution, we have the expression below

$$
S_n(x) = \frac{1+x}{1 + x - n} \sum_{k=0}^{n} (-1)^{k} \frac{\binom{1+x-n}{k} \binom{x-n-k}{n-k}}{\binom{n+k}{k}} S_1(n+k, k),
$$

which confirms the second formula. □

There exists a known, but slightly different formula, as shown below:

$$
\begin{aligned}
S_n(x) &= \binom{x-n}{n} \sum_{k=0}^{n} (-1)^{k} \frac{x-2n}{x-n-k} \frac{n!^2 S_1(n+k+1, k+1)}{(n+k)!(n-k)!} \\
&= \sum_{k=0}^{n} (-1)^{k} \frac{\binom{x-n}{k} \binom{x-n-k-1}{n-k}}{\binom{n+k}{k}} S_1(n+k+1, k+1).
\end{aligned}
$$

The above formula is located in https://en.wikipedia.org/wiki/Stirling_polynomials (accessed on 1 November 2022) and simpler than the double sum expression that appears in [27]. Further expressions of a similar type can be found in [65,66].

The above formula can be shown by constructing the Lagrange interpolating polynomial for $S_n(x)$ at $\{n+k\}_{k=0}^{n}$ as follows

$$
S_n(x) = \sum_{k=0}^{n} \Lambda_k \frac{\langle x - n \rangle_n}{x - n - k},
$$

where the connection coefficients are determined by

$$
\Lambda_k = \frac{S_n(n+k)}{\prod_{\substack{\imath=0 \\ \imath \neq k}}^{n} (k - \imath)} = (-1)^{n-k} \frac{S_n(n+k)}{k!(n-k)!}
$$

and

$$
S_n(n+k) = (-1)^{n} S_1(n+k+1, k+1) \Big/ \binom{n+k}{n}.
$$

Finally, we record typical properties of the Stirling polynomials.

- Extreme values:

$$S_n(n) = n! \quad \text{and} \quad S_n(0) = (-1)^n B_n,$$
$$S_n(1) = (-1)^{n-1}\big\{(n-1)B_n + nB_{n-1}\big\}.$$

- Stirling numbers:

$$S_n(m) = (-1)^n S_1(1+m, 1+m-n)\Big/\binom{m}{n},$$
$$S_n(-m) = S_2(n+m-1, m-1)\Big/\binom{-m}{n}.$$

- Binomial convolution:

$$S_n(x+y-1) = \sum_{k=0}^{n} \binom{n}{k} S_k(x-1)S_{n-k}(y-1).$$

- Linear relations:

$$\binom{m+n}{n} S_n(x+m) = \sum_{k=0}^{n} \binom{m+n}{k} S_1(m, m-n+k)S_k(x),$$
$$\binom{m+n}{n} S_n(x-m) = \sum_{k=0}^{n} (-1)^{n-k}\binom{m+n}{k} S_2(n-k+m, m)S_k(x).$$

**Funding:** This research received no external funding.

**Data Availability Statement:** Not applicable.

**Acknowledgments:** The author is sincerely grateful to three anonymous referees for their careful reading, critical comments, and valuable suggestions that improved the manuscript substantially during the revision. Special thanks go to Nadia Na Li for having made a thoroughgoing check and detected several errors and typos on the initial version of this manuscript.

**Conflicts of Interest:** The author declares no conflict of interest.

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
