# Peer review of "Reciprocal Symmetry via Inverse Series Pairs"

_symmetry, doi:10.3390/sym15051086_

Round 1

Reviewer 1 Report

In this article, the author discusses various noteworthy number/polynomial sequence pairings linked by symmetric reciprocities. An overview of explicit expressions, recurrence relations, orthogonality relations, convolution sums, and reciprocal formulae for several typical classical sequences, such as binomial coefficients, Stirling numbers, and Bernoulli/Euler numbers and polynomials, will be presented systematically using reciprocal function pairs and formal power series expansions.

Their method of computing is novel and their technique is fascinating. To my knowledge, results are new and worth publication as there are addressing very important classes of series.

·       Add recent literature related to INVERSE SERIES PAIRS.

·       Minor spelling check required.

·       Conclusion section must add some possible future directions and open problems in this direction.

·       Please add an introduction to your proposed work.

·       Add keyword list after the abstract, please add abstract hading.

·       Please add the graphical difference between the given various series in Section 1.

·       Please add the graphical difference of given polynomials in Section 7.

Author Response

see uploaded pdf-file

Reviewer 2 Report

The author applied reciprocal function and formal power series to some classical sequences to present explicit expressions, recurrence relations, orthogonality relations, convolution sums and reciprocal formulae. 

The results are very interesting. However I have an issue with the presentation of the paper as it is written in  narrative writing. In the whole article there is not one theorem!

I would suggest that the results need to be rewritten in Theorem\Proof format.

There are some minor corrections:

1- In page 2, definition of \sigma_n(m). I think the author meant the number of partitions of m into n parts not the set of partitions. 

2- In page 6, the author needs to define the Pochhammer symbol (n-m)_m as well as 4_F_3 series.

3- In page 9, in the last two formulas, is m conquerant to 2 mod 1? As i do not get the condition for \chi(m).  

4-  In page 21, second line, missing square bracket. 

5- In page 26, end of proof symbol is missing. 

6- In page 27, instead of writing wikipedia link, it more scientific to find an appropriate reference as the information in wekipidia is editable by anyone. 

Author Response

see uploaded pdf-file

Reviewer 3 Report

See attachement.

English is adequatee and only  minor corrections listed in the attached report are required.

Author Response

see uploaded pdf-file
